# Enantiocomplementary Bioreduction of 1-(Arylsulfanyl)propan-2-ones

**DOI:** 10.3390/molecules29163858

**Published:** 2024-08-15

**Authors:** Emese Sándor, Pál Csuka, László Poppe, József Nagy

**Affiliations:** Department of Organic Chemistry and Technology, Faculty of Chemical Technology and Biotechnology, Budapest University of Technology and Economics, Műegyetem rkp. 3, H-1111 Budapest, Hungary; emese.sandor@edu.bme.hu (E.S.); csuka.pal@vbk.bme.hu (P.C.)

**Keywords:** ketoreductase, whole-cell bioreduction, yeast, thiophenol derivatives, alcohol dehydrogenase

## Abstract

This study explored the enantiocomplementary bioreduction of substituted 1-(arylsulfanyl)propan-2-ones in batch mode using four wild-type yeast strains and two different recombinant alcohol dehydrogenases from *Lactobacillus kefir* and *Rhodococcus aetherivorans.* The selected yeast strains and recombinant alcohol dehydrogenases as whole-cell biocatalysts resulted in the corresponding 1-(arylsulfanyl)propan-2-ols with moderate to excellent conversions (60–99%) and high selectivities (ee > 95%). The best bioreductions—in terms of conversion (>90%) and enantiomeric excess (>99% ee)—at preparative scale resulted in the expected chiral alcohols with similar conversion and selectivity to the screening reactions.

## 1. Introduction

Enantiopure chiral compounds are important and useful intermediates and building blocks in the synthesis of drugs and their precursors. Sulfur containing chiral secondary alcohols, like β-hydroxysulfides, are relevant intermediates in the syntheses of naturally occurring spiroketal pheromones [1], chiral oxiranes [2], thiiranes [3], tetrahydrofurans [4], and 4-acetoxyazetidinone [5]. Moreover, bicalutamide and their analogs, being anti-cancer drugs, contain β-hydroxysulfide linker unit [6]. In addition, sulfides can be easily oxidized to the corresponding sulfoxides and sulfones. The enantiomers of the latter compounds can be produced by enzymatic and chemo-enzymatic kinetic resolution [7]. These resolutions can be performed with *Candida antarctica* lipase B and *Humicola lanuginosa* lipase [8,9,10]. Bioreductions of arylsulfonyl-, arylsulfinyl- and arylsulfanylketones by baker’s yeast were investigated [10]. Due to the additional center of asymmetry at *S*-atom, all four stereoisomers of the 1-arylsulfinylpropan-2-ols were prepared. Redox biotransformations of arylsulfanylketones with *Helminthosporium* sp., *Mortierella isabellina*, and *Rhodococcus erythropolis* resulted in arylsulfonyl-, arylsulfinyl- and arylsulfanyl-alcohols [11]. Notable are the elegant chemo-biocatalytic cascade reactions utilizing purified (*S*)- or (*R*)-selective ketoreductases (KREDs) and NADH-supplemented systems to produce the (*S*)- or (*R*)-1-(arylsulfanyl)propan-2-ols starting from 1-chloropropan-2-ol or chloroacetone and tiophenols [12]. Thus, in contemporary research, there is a broad interest in stereosynthesis of chiral-sulfur-containing alcohols.

Since KREDs are important elements of the toolbox of industrial biocatalysis [13,14,15,16], various forms of (*S*)- and (*R*)-selective KREDs (named also as alcohol dehydrogenases, ADHs) were explored for the production of a representative set of 1-[(substituted)arylsulfanyl]propan-2-ols. The best-performing KRED-catalyzed processes were scaled up to give the expected chiral alcohols at preparative scale.

The alcohol dehydrogenase from *Rhodococcus aertherivorans* (RaADH; also known as *Lactobacillus brevis* ADH, LbADH) is an excellent biocatalyst with a broad range of ketones—such as variously substituted aromatic ketones and aliphatic polyketides—accepted as substrate to form (*S*)-alcohols with high enantiotopic selectivity [17,18,19,20].

The alcohol dehydrogenase from *L. kefir* (LkADH) is another longtime-used biocatalyst to produce (*R*)-alcohols. The broad substrate scope renders this enzyme as an efficient biocatalyst with ketoreductase activity. LkADH can efficiently produce aromatic, cyclic, polycyclic, and aliphatic (*R*)-alcohols [21,22].

The ketoreduction by whole-cell biocatalysts is an efficient technology because no external cofactor supplementation is needed, and the cofactor regeneration required for the biocatalysis happens within the cells. For an efficient (high-conversion) ketoreduction, an appropriate cofactor regeneration strategy is required. For the NAD(P)H regeneration in the KRED-catalyzed bioreductions, isopropanol, glucose, or lactose can serve as co-substrate providing the hydrogen to reduce the forming NAD(P)^+^. The nature of the co-substrate depends on the enzyme system applied for the cofactor regeneration.

Wild-type yeast strains are useful biocatalysts in biosynthetic ketoreductions because the regeneration of the intrinsic cofactor is highly effective in their whole cells. The most frequently used strains are the *Pichia* (*Pichia carsonii*), *Candida* (*Candida parapsilosis*, *Candida albicans*, *Candida norvegica*), and *Lodderomyces* (*Lodderomyces elongisporus*) strains. The KREDs in these strains usually follow the Prelog’s rule [23] and catalyze the formation of (*S*)-alcohols in high enantiomeric purity from a broad range of ketones [24,25,26,27,28,29,30].

Because the usefulness of various (*S*)-1-(arylsulfanyl)propan-2-ols was already indicated but no single-step bioreductions leading to (*R*)-1-(arylsulfanyl)propan-2-ols from the corresponding ketones were found, our goal was to develop high-efficiency methods for the preparation of both enantiomeric forms of 1-(arylsulfanyl)propan-2-ols using enantiocomplementary KREDs in their whole-cell forms.

## 2. Results and Discussion

In this study, four yeast strains from the Witaria culture collection [Budapest, Hungary: *P. carsonii* (WY1), *L. elongisporus* (WY2), *C. norvegica* (WY4), *C. parapsilosis* (WY12)] and recombinant enzymes as whole cells [*L. kefir* (*R*)-alcohol dehydrogenase (LkADH) and *R. aetherivorans* (*S*)-alcohol dehydrogenase (RaADH)] were investigated as biocatalysts in the bioreduction in a series of ketones with variously substituted arylsulfanyl moieties {**2a**: 1-(phenylsulfanyl)propan-2-one, **2b**: 1-[(4-methylphenyl)sulfanyl]propan-2-one, **2c**: 1-[(4-chlorophenyl)sulfanyl]propan-2-one, **2d**: 1-[(2,5-dichlorophenyl)sulfanyl]propan-2-one, **2e**: 1-[(4-methoxyphenyl)sulfanyl]-propan-2-one} (Figure 1). 

The ketones with different substituents attached to the aromatic ring (**2a**–**e**) were prepared by alkylation of substituted thiophenols (**3a**–**e**) with chloroacetone (**4**) (Figure 1). Racemic alcohols ((±)-**1a**–**e**) for setting up enantioselective GC were prepared by the sodium borohydride reduction of the corresponding ketones **2a**–**e**.

The conversions of bioreductions were followed by GC analysis of samples taken from the reaction mixtures at 2, 4, 8, and 24 h reaction times (Figure 1). High enantiotopic selectivities were observed in all bioreductions, resulting in virtually enantiopure alcohols [ee > 99%; for (*S*)-**1a**–**e** with *P*. *carsonii* (WY1), *L. elongisporus* (WY2), *C. norvegica* (WY4), *C. parapsilosis* (WY12) and *R. aetherivorans* (*S*)-alcohol dehydrogenase (ReADH), and for (*R*)-**1a**–**e** with *L. kefir* (*R*)-alcohol dehydrogenase (LkADH)].

The highest activities were observed with the WY1, WY2, and WY12 yeast strains in the cases of ketones **2a**–**c** (80–99% conversions at 2 h reaction time). In the case of WY1, bioreduction of all ketones **2a**–**e** resulted in products (*S*)-**1a**–**e** in good conversions with excellent selectivity (>99% ee) in 8 h reaction time (Figure 1a). The bioreduction of **2d** was the slowest due to the highest steric hindrance of the aryl group bearing two chlorine substituents. In the case of WY2, the products formed with excellent selectivity (>99% ee) from **2a**–**d** in good conversions within 4 h, while the conversion of **2e** remained below 80% at 24 h (Figure 1b). WY4 was the least efficient but still quite selective biocatalyst among the investigated yeasts, producing (*S*)-**1a**–**e** (>99% ee) with the lowest conversions (Figure 1c). The best results were obtained with WY12, performing the bioreductions with excellent conversion and enantioselectivity (c > 99%; ee > 99%) even within 2 h (Figure 1d).

The bioreductions of ketones **2a**–**e** with enantiocomplementary recombinant alcohol dehydrogenases—the *L. kefir* (*R*)-alcohol dehydrogenase (LkADH) and *R. aetherivorans* (*S*)-alcohol dehydrogenase (RaADH)—were investigated as whole-cell biocatalysis. In the case of LkADH with (*R*)-(i.e., anti-Prelog) selectivity, excellent conversion and enantiotopic selectivity (c > 99%, ee > 99%) were obtained in the reduction of ketones **2a**–**c** and **2e** (Figure 1e). The steric hindrance by the dichloro-substituted phenyl moiety rendered the transformation of ketone **2d** slower. The workup process of this biotransformation was also the most challenging due to the highest affinity of the hydrophobic alcohol (*R*)-**1d** to the cell constituents during extraction from the reaction mixture. Reactions with RaADH with the usual Prelog (*S*)-selectivity proceeded with lower conversions but also high enantioselectivities. Moreover, the non-complete conversions at longer reaction time possibly indicated product inhibition even at low substrate concentration (~10 mM)—except for the unsubstituted compound **2a** with reasonable conversion in 4 h (Figure 1f).

Next, the best-performing bioreductions in terms of conversion of ketones **2a**–**e** leading either to the (*S*)- or to the (*R*)-enantiomer of the alcohols (**1a**–**e**) (WY12 and LkADH, respectively) were performed at preparative scale to isolate and characterize the enantiopure products (Table 1).

The most challenging aspect of the biotransformations was the emulsion formation during the extraction. Extraction was not feasible without previous centrifugation due to intense emulsion formation. Increasing the temperature of the centrifugation from 4 °C to 24 °C improved the efficiency of this step. Although extraction with the green solvent methyl tert-butyl ether would be desirable, the extraction by diethyl ether proved to be more efficient. The nature of forming alcohols (and the trace of residual ketones) had also a significant effect on the emulsion formations. The least intensive emulsion formation was observed during the working up of the reaction mixture of (*S*)- and (*R*)-**1a** with unsubstituted phenyl ring, leading to the highest preparative yields (entries 1 and 6, Table 1). In case of the other biotransformations, the handling of bioreactions with LkADH (entries 7–10) was less difficult than with the ones performed with the WY12 yeast whole cells (entries 2–5), resulting in higher isolated yields. All the isolated (*S*)- and (*R*)-products were virtually enantiopure (ee > 99%, determined by GC), having exactly opposite optical rotations.

## 3. Materials and Methods

### 3.1. Chemicals

Thiophenol derivatives, the reagents, the solvents, tryptone and yeast extract were purchased from Merck KGaA (Darmstadt, Germany) and Molar Chemicals (Halásztelek, Hungary). 

### 3.2. Biocatalysts

#### 3.2.1. Lyophilized Yeast Whole-Cell Preparations

The lyophilized whole-cell yeasts in this study originated from the Witaria (Budapest, Hungary) strain culture collection (*P. carsonii* (WY1), *L. elongisporus* (WY2), *C. norvegica* (WY4) and *C. parapsilosis* (WY12)) [31,32,33]. The fermentation and lyophilization of the yeast strains were described earlier [31,32].

#### 3.2.2. Recombinant Alcohol Dehydrogenase Whole-Cell Preparations

The recombinant KREDS used in our study (the (*S*)-selective *R. aertherivorans* alcohol dehydrogenase (RaADH) [17,18,19,20] and the (*R*)-selective *L. kefir* alcohol dehydrogenase (LkADH) [21,22]) were comprehensively investigated on similar substrates, including optimization of the bioreductions. The codon-optimized genes of LkADH (Uniprot ID: Q6WVP7) and RaADH (Uniprot ID: N1MBR5) were purchased from GenScript (Rijswijk, the Netherlands). The genes were placed into a pET19b vector with NdeI and BamHI restriction enzyme sites. The plasmids were transformed into the *Escherichia coli* BL21 strain with the heat shock method. The transformed cells were growing in 5 mL of LB media (tryptone, 50 mg; yeast extract, 25 mg; NaCl, 50 mg; pH = 7.0) supplemented with carbenicillin (50 µg mL^−1^) overnight at 37 °C at 180 rpm. For protein expression, 250 mL of autoinduction media (Na_2_HPO_4_, 1.5 g; NaH_2_PO_4_, 0.75 g; tryptone, 5 g; yeast extract, 1.25 g; NaCl, 1.25 g) and 10 mL of carbon supplementing solution (glycerol, 1.89 g; glucose, 125 mg; lactose, 500 mg) was inoculated with 1 mL of overnight culture. The cultures were shaken at 37 °C at 180 rpm for 16 h (reaching typically OD_600_~1). The cells were separated with centrifugation at 4 °C at 2500 g for 20 min. After separation, the cells were resuspended in 10 mL of phosphate buffer (pH = 7.5, 50 mM) and centrifuged again with the previous settings. The cell pellet (typically ~1.6 g wet weight) was resuspended in 10 mL of phosphate buffer (pH = 7.5, 50 mM) and used directly for the biocatalysis.

### 3.3. Analytical Methods

The NMR spectra were recorded in CDCl_3_ and DMSO-d_6_ on a DRX-300 spectrometer (Bruker; Karlsruhe, Germany) operating at 500 MHz for ^1^H, and 125 MHz for ^13^C, and signals are given in ppm on the δ scale. Infrared spectra were recorded on a ALPHA FT-IR spectrometer (Bruker; Karlsruhe, Germany), and wavenumbers of bands are listed in cm^−1^. Optical rotation was measured on Model 241 polarimeter (Perkin-Elmer; Shelton, CT, USA) at the D-line of sodium. The conversion from **2a**–**e** and the enantiomeric excess of the formed alcohol (*S*)-**1a**–**e** or (*R*)-**1a**–**e** in the bioreductions were determined by gas chromatography (GC) using an 4890 GC (Agilent; Santa Clara, CA, USA) equipped with an FID detector and a Hydrodex β-6TBDM column [25 m × 0.25 mm × 0.25 μm film with heptakis-(2,3-di-*O*-methyl-6-*O*-*t*-butyldimethylsilyl)-β-cyclodextrin; Macherey & Nagel (Düren, Germany); H_2_ carrier gas (injector: 250 °C, detector: 250 °C, head pressure: 12 psi, split ratio: 50:1)]. The temperature programs and retention times are included in the Appendix A.

#### 3.3.1. Sampling for Determination of Conversion in Bioreductions by GC

Samples taken from the reaction mixture (50 μL) were extracted with MTBE (700 μL). The organic layer was dried over sodium sulfate (70 mg, for ~5 min), and the decanted MTBE-extracts were analyzed directly. The molar response factors for FID detection are included in the Appendix A.

#### 3.3.2. Sampling for Determination of Conversion in Bioreductions by GC

For the detection of the enantiomeric excess, samples taken from the reaction mixtures (500 μL) were extracted with MTBE (700 μL). After drying the organic layer over sodium sulfate 70 mg, for ~5 min), trifluoroacetic anhydride (30 mg) was added to the decanted solution and the acylation was conducted at 30 °C for 12 h to give (*R*)- and (*S*)-**5a**–**c**,**e**. The excess of the acylating agent was quenched with water (25 mg, for ~30 min), the organic layer was dried over dry sodium carbonate (~100 mg), and the decanted MTBE extracts were analyzed directly. In the case of (*R*)- and (*S*)-**1d**, derivatization was not necessary to separate the enantiomers by GC.

### 3.4. Chemical Synthesis of the Substrates and Reference Compounds

#### 3.4.1. Synthesis of 1-(Arylsulfanyl)propan-2-ones **2a**–**e**

To a solution of the corresponding benzenethiol (**3a**–**e**, 10 mmol) in acetonitrile (40 mL) at room temperature was added anhydrous triethylamine (25 mmol, 2.5 eq.). After cooling to 0 °C, we added dropwise to the stirred arylthiol solution a solution of chloroacetone (**4**, 20 mmol, 2 eq., in 10 mL of acetonitrile; ~30 min). The resulting mixture was allowed to reach room temperature, and then it was stirred at 40 °C until completion of the reaction (~8 h). After evaporating off the solvent from the reaction mixture, water (40 mL) was added to the residue and extraction was performed with dichloromethane (3 × 40 mL). The combined organic phases were washed with water (20 mL) and brine (20 mL), dried over sodium sulfate, and concentrated in vacuum. The residue was purified by column chromatography over silica gel (eluent: hexane/ethyl acetate 20:1 to 10:1) to give the desired ketone **2a**–**e**.

1-(Phenylsulfanyl)propan-2-one, **2a**: White crystals (66% yield); Rf = 0.45 (7:1 hexane/EtOAc); melting point: 31–35 °C; ^1^H NMR (CDCl_3_, 500 MHz) δ 7.30–7.25 (m, 2H, 2′,6′-H), 7.24–7.20 (m, 2H, 3′,5′-H), 7.17–7.12 (m, 1H, 4′-H), 3.60 (s, 2H, CH_2_), 2.21 (s, 3H, CH_3_); ^13^C NMR (CDCl_3_, 125 MHz) δ 203.5 (C=O), 134.6 (Ar-C), 129.5 (Ar-C), 126.9 (Ar-C), 44.7 (CH_2_), 28.0 (CH_3_); IR (KBr): ν (cm^−1^) 1708 (C=O), 1479 (aromatic C=C). The spectral data agreed with the published IR and ^1^H-NMR [34], as well as ^13^C-NMR [35] data. 

1-[(4-Methylphenyl)sulfanyl]propan-2-one, **2b**: Yellowish oil (70% yield); R_f_ = 0.5 (7:1 hexane/EtOAc); ^1^H NMR (DMSO-d_6_, 500 MHz) δ 7.22 (d, *J* = 8 Hz, 2H, Ar-H); 7.13 (d, *J* = 8 Hz, 2H, Ar-H); 3.91 (s, 2H, CH_2_); 2.26 (s, 3H, CH_3_); 2.19 (s, 3H, Ar-CH_3_); ^13^C NMR (DMSO-d_6_, 125 MHz) δ 203.7 (C=O), 136.2 (Ar-C), 131.9 (Ar-C), 130.2 (Ar-C), 129.3 (Ar-C), 44.19 (CH_2_), 28.7 (CH_3_), 20.9 (Ar-CH_3_); IR (KBr): ν (cm^−1^) 1705 (C=O), 1492 (aromatic C=C). The spectral data agreed with the published IR [36] as well as the ^1^H-NMR and ^13^C-NMR [34] data.

1-[(4-Chlorophenyl)sulfanyl]propan-2-one, **2c**: White crystals (53% yield); R_f_ = 0.2 (10:1 hexane/EtOAc); melting point: 28–30 °C; ^1^H NMR (CDCl_3_, 500 MHz) δ 7.19 (m, 4H, Ar-H); 3.57 (s, 2H, CH_2_); 2.20 (s, 3H, CH_3_); ^13^C NMR (CDCl_3_, 125 MHz) δ 203.0 (C=O), 133.1 (Ar-C), 131.0 (Ar-C), 129.3 (Ar-C), 44.7 (CH_2_), 28.0 (CH_3_); IR (KBr): ν (cm^−1^) 1705 (C=O), 1492 (aromatic C=C) 809 (Cl-C). The spectral data agreed with the published IR [34], as well as the ^1^H-NMR and ^13^C-NMR [35] data.

1-[(2,5-Dichlorophenyl)sulfanyl]propan-2-one, **2d**: White crystals (46% yield); Rf = 0.35 (10:1 hexane/EtOAc); melting point: 92–94 °C; ^1^H NMR (CDCl_3_, 500 MHz) δ 7.29 (d, *J* = 8.5 Hz, 1H, Ar-H), 7.24 (d, *J* = 2.5, 1H, Ar-H); 7.11 (dd, *J* = 8.5 Hz, 2.5 Hz, 1H, Ar-H), 3.74 (s, 2H, CH_2_); 2.32 (s, 3H, CH_3_) ^13^C NMR (CDCl_3_, 125 MHz) δ 202.2 (C=O), 136.1 (Ar-C), 133.2 (Ar-C), 131.7 (Ar-C), 130.6 (Ar-C), 128.4 (Ar-C), 127.4 (Ar-C), 43.2 (CH_2_), 28.2 (CH_3_); IR (KBr): ν (cm^−1^) 1693 (C=O), 1447 (aromatic C=C), 800 (Cl-C).

1-[(4-Methoxyphenyl)sulfanyl]propan-2-one, **2e**: Yellow oil (67% yield); R_f_ = 0.25 (10:1 hexane/EtOAc); ^1^H NMR (CDCl_3_, 500 MHz) δ 7.35 (d, *J* = 8.5 Hz, 2H, Ar-H); 6.83 (d, *J* = 8.5 Hz, 2H, Ar-H); 3.79 (s, 3H, O-CH_3_); 3.55 (s, 2H, CH_2_); 2.26 (s, 3H, CH_3_); ^13^C NMR (CDCl_3_, 125 MHz) δ 203.6 (C=O), 159.6 (Ar-C), 133.6 (Ar-C), 124.5 (Ar-C), 114.8 (Ar-C), 55.3 (O-CH_3_), 46.6 (CH_2_), 28.1 (CH_3_); IR (KBr): ν (cm^−1^) 1703 (C=O), 1493 (aromatic C=C); 1243 (C–O). The spectral data agreed with the published ^1^H-NMR and ^13^C-NMR [35] data.

#### 3.4.2. Synthesis of Racemic 1-(Arylsulfanyl)propan-2-ols (±)-**1a**–**e**

Sodium borohydride (2.35 mmol, 1.3 eq.) was added slowly to a 30 mL methanol solution of the corresponding ketone (**2a**–**e**) (1.8 mmol, 1 eq.) at 0 °C. The reaction mixture was stirred for 30 min at 0 °C then quenched with 440 μL of concentrated acetic acid. The reaction mixture was evaporated, diluted with water (40 mL), and extracted with diethyl ether (3 × 40 mL). The combined organic phases were washed with saturated sodium hydrogen carbonate solution (30 mL) and water (30 mL), dried over sodium sulfate, and concentrated in vacuum. The product was dried under vacuum overnight.

1-(Phenylsulfanyl)propan-2-ol, (±)-**1a:** Colorless liquid (80% yield), R_f_ = 0.45 (5:1 hexane/EtOAc); ^1^H NMR (CDCl_3_, 500 MHz) δ 7.41–7.37 (m, 2H, 2′,6′-H), 7.32–7.27 (m, 2H, 3′,5′-H), 7.26–7.21 (m, 1H, 4′-H), 3.84 (dqd, *J* = 8.5, 6.2, 3.7 Hz, 1H, 2-CH), 3.11 (dd, *J* = 13.7, 3.7 Hz, 1H, 3-CH), 2.85 (dd, *J* = 13.7, 8.5 Hz, 1H, 3-CH), 1.27 (d, *J* = 6.2 Hz, 3H, CH_3_); ^13^C NMR (CDCl_3_, 125 MHz) δ 135.2 (Ar-C), 130.1 (Ar-C), 129.1 (Ar-C), 126.7 (Ar-C), 65.6 (C-OH), 43.6 (CH_2_), 21.9 (CH_3_); IR (KBr): ν (cm^−1^) 3371 (OH), 1438 (aromatic C=C). The spectral data agreed with the published IR and the ^1^H-NMR [37], as well as with the ^13^C-NMR [38] data.

1-[(4-Methylphenyl)sulfanyl]propan-2-ol, (±)-**1b:** Yellowish liquid (89% yield), R_f_ = 0.4 (5:1 hexane/EtOAc); ^1^H NMR (CDCl_3_, 500 MHz) δ 7.31 (d, *J* = 8 Hz, 1H, Ar-H), 7.11 (d, *J* = 8 Hz, 1H, Ar-H), 3.85–3.78 (m, 1H, CH), 3.06 (dd, *J* = 13.5 Hz, 3.5 Hz, 1H, CH_2_), 2.79 (dd, *J* =13.5 Hz, 8 Hz, 1H, CH_2_), 2.49 (s, 1H, OH), 2.32 (s, 3H, CH_3_), 1.25 (d, *J* = 6 Hz, 3H, CH_3_); ^13^C NMR (CDCl_3_, 125 MHz) δ 136.9 (Ar-C), 131.3 (Ar-C), 130.9 (Ar-C), 129.8 (Ar-C), 65.4 (C-OH), 44.4 (CH_2_), 21.8 (CH_3_), 21.0 (Ar-CH_3_); IR (KBr): ν (cm^−1^) 3372 (OH), 1480 (aromatic C=C). The spectral data agreed with the published IR and the ^1^H-NMR [39], as well as the ^13^C-NMR [40].

1-[(4-Chlorophenyl)sulfanyl]propan-2-ol, (±)-**1c**: Yellow liquid (90% yield), R_f_ = 0.35 (5:1 hexane/EtOAc); ^1^H NMR (CDCl_3_, 500 MHz) δ 7.34 (d, *J* = 8.5 Hz, 2H, Ar-H), 7.28 (d, *J* = 8.5 Hz, 2H, Ar-H), 3.90–3.84 (m, 1H, CH), 3.09 (dd, *J*= 13.5 Hz, 3.5 Hz, 1H, CH_2_), 2.87 (dd, *J*= 13.5 Hz, 8.5Hz, 1H, CH_2_), 2.37 (s, 1H, OH), 1.29 (d, *J* = 6 Hz, 3H, CH_3_); ^13^C NMR (CDCl_3_, 125 MHz) δ 133.9 (Ar-C); 132.7 (Ar-C), 131.4 (Ar-C), 129.2 (Ar-C), 65.6 (C-OH), 43.8 (CH_2_), 22.0 (CH_3_), IR (KBr): ν (cm^−1^) 3363 (OH) 1472 (aromatic C=C), 801 (C-Cl). The spectral data agreed with the published ^1^H-NMR [39].

1-[(2,5-Dichlorophenyl)sulfanyl]propan-2-ol, (±)-**1d:** White solid (87% yield), R_f_ = 0.3 (5:1 hexane/EtOAc); melting point: 77–79 °C ^1^H NMR (CDCl_3_, 500 MHz) δ 7.30 (d, *J* = 2.5 Hz, 1H, Ar-H); 7.29 (s, 1H, Ar-H); 7.10 (dd, *J* = 8.5, 2.0 Hz; 1H, Ar-H); 3.98–3.92 (m, 1H, CH); 3.11 (dd, *J* = 13.5, 4 Hz, 1H, CH_2_), 2.93 (dd, *J* = 13.5, 8.5Hz, 1H, CH_2_), 2.27 (s, 1H, OH), 1.33 (d, *J* = 6.5 Hz, 3H, CH_3_). ^13^C NMR (CDCl_3_, 125 MHz) δ 137.1 (Ar-C), 133.1 (Ar-C), 132.2 (Ar-C), 130.66 (Ar-C), 128.7 (Ar-C), 127.1 (Ar-C), 65.77 (C-OH), 42.1 (CH_2_), 22.3 (CH_3_). IR (KBr): ν (cm^−1^) 3216 (OH) 1447 (aromatic C=C), 798 (C-Cl).

1-[(4-Methoxyphenyl)sulfany]propan-2-ol, (±)-**1e:** Colorless liquid (85% yield), R_f_ = 0.25 (5:1 hexane/EtOAc); ^1^H NMR (CDCl_3_, 500 MHz) δ 7.38 (d, *J* = 9 Hz, 2H, Ar-H), 6.84 (d, *J* = 9 Hz, 2H, Ar-H), 3.79 (s, 3H, O-CH_3_), 3.78–3.73 (m, 1H, CH), 2.98 (dd, *J* = 13.5, 3.5 Hz, 1H, CH_2_), 2.72 (dd, *J* = 13.5, 8.5 Hz, 1H, CH_2_), 2.54 (s, 1H, OH), 1.22 (d, *J* = 6.5 Hz, 3H, CH_3_); ^13^C NMR (CDCl_3_, 125 MHz) δ 159.3 (Ar-C), 133.8 (Ar-C), 125.1 (Ar-C), 114.7 (Ar-C), 65.3 (C-OH), 55.4 (O-CH_3_), 45.7 (CH_2_), 21.7 (CH_3_). IR (KBr): ν (cm^−1^) 3405 (OH) 1492 (aromatic C=C).

### 3.5. Bioreductions of 1-(Arylsulfanyl)propan-2-ones (***2a***–***e***) to (S)-1-(Arylsulfanyl)propan-2-ols (S)-***1a***–***e***

The conditions optimized in our previous study for similar bioreductions with lyophilized yeast cells [33] were applied. Thus, yeast cells (30 mg) were suspended in sodium phosphate buffer (850 μL, pH = 7, 50 mM). To the suspension were added 2-propanol (50 μL) and a solution of the substrate ketone (**2a**–**e**) (100 μL, 2 mg mL^−1^ in DMSO). The resulting mixture was shaken at 350 rpm for 24 h at room temperature. After 24 h, the mixture was extracted with MTBE (3 × 1 mL) and dried over sodium sulfate. 

### 3.6. Bioreduction of 1-(Arylsulfanyl)propan-2-ones (***2a***–***e***) to (S)- or (R)-1-(Arylsulfanyl)propan-2-ols (S)- or (R)-***1a***–***e***

The conditions applied here were similar as optimized earlier for reaction with recombinant whole-cell LkADS [19] and RaADH [41], except for the reduced amount of 2-propanol (5 *v*/*v*%) due to the larger amount of DMSO (10 *v*/*v*%) applied in the present study to enhance the substrate solubility. Thus, alcohol dehydrogenase whole-cell suspension (200 μL, see Section 3.2.2 for its preparation) was added to sodium phosphate buffer (650 μL, pH = 7, 50 mM). To the suspension were added 2-propanol (50 μL) and a solution of the substrate ketone (**2a**–**e**) (100 μL, 2 mg mL^−1^ in DMSO). The resulting mixture was shaken at 350 rpm for 24 h at room temperature. After 24 h, the mixture was extracted with MTBE (3 × 1 mL) and dried over sodium sulfate. 

### 3.7. Bioreduction of 1-(Arylsulfanyl)propan-2-ones (***2a***–***e***) to (S)-1-(Arylsulfanyl)propan-2-ols (S)-***1a***–***e*** at Preparative Scale

Lyophilized WY12 yeast cells (1 g) were suspended in sodium phosphate buffer (42.5 mL, pH = 7, 50 mM). To the resulting suspension were added 2-propanol (2.5 mL) and the substrate (**2a**–**e**, 100 mg in 5 mL of DMSO). The resulting mixture was shaken at 350 rpm for 24 h at room temperature. After centrifugation for 10 min at ~4000× *g*, 24 °C, the supernatant was extracted with diethyl ether (3 × 50 mL), and the combined organic layers were dried over sodium sulfate and evaporated under reduced pressure. The crude residue was purified by column chromatography over silica gel (eluent: hexane/ethyl acetate 10:1 to 7:1) to give the desired alcohols (*S*)-**1a**–**e**.

(2*S*)-1-(Phenylsulfanyl)propan-2-ol (*S*)-**1a:** Colorless liquid (75% yield), ^1^H NMR (CDCl_3_, 500 MHz) δ 7.41–7.37 (m, 2H, 2′,6′-H), 7.32–7.27 (m, 2H, 3′,5′-H), 7.26–7.21 (m, 1H, 4′-H), 3.84 (dqd, *J* = 8.9, 6.2, 3.7 Hz, 1H, 2-CH), 3.11 (dd, *J* = 13.7, 3.7 Hz, 1H, 3-CH), 2.85 (dd, *J* = 13.7, 8.5 Hz, 1H, 3-CH), 1.27 (d, *J* = 6.2 Hz, 3H, CH_3_); ^13^C NMR (CDCl_3_, 125 MHz) δ 135.2 (Ar-C), 130.1 (Ar-C), 129.1 (Ar-C), 126.7 (Ar-C), 65.6 (C-OH), 43.6 (CH_2_), 21.9 (CH_3_). The spectral data agreed with the published ^1^H-NMR, ^13^C-NMR [11] data. 

(2*S*)-1-[(4-Methylphenyl)sulfanyl]propan-2-ol (*S*)-**1b:** Yellowish liquid (36% yield), ^1^H NMR (CDCl_3_, 500 MHz) δ 7.31 (d, *J* = 8 Hz, 1H, Ar-H), 7.11 (d, *J* = 8 Hz, 1H, Ar-H), 3.85–3.78 (m, 1H, CH), 3.06 (dd, *J* = 13.5 Hz, 3.5 Hz, 1H, CH_2_), 2.79 (dd, *J* = 13.5 Hz, 8 Hz, 1H, CH_2_), 2.49 (s, 1H, OH), 2.32 (s, 3H, CH_3_), 1.25 (d, *J* = 6 Hz, 3H, CH_3_); ^13^C NMR (CDCl_3_, 125 MHz) δ 136.9 (Ar-C), 131.3 (Ar-C), 130.9 (Ar-C), 129.8 (Ar-C), 65.4 (C-OH), 44.4 (CH_2_), 21.8 (CH_3_), 21.0 (Ar-CH_3_). The spectral data agreed with the published ^1^H-NMR [10] data.

(2*S*)-1-[(4-Chlorophenyl)sulfanyl]propan-2-ol (*S*)-**1c:** Yellow liquid (58% yield), ^1^H NMR (CDCl_3_, 500 MHz) δ 7.34 (d, *J* = 8.5 Hz, 2H, Ar-H), 7.28 (d, *J* = 8.5 Hz, 2H, Ar-H), 3.90–3.84 (m, 1H, CH), 3.09 (dd, *J* = 13.5 Hz, 3.5 Hz, 1H, CH_2_), 2.87 (dd, *J* = 13.5 Hz, 8.5 Hz, 1H, CH_2_), 2.37 (s, 1H, OH), 1.29 (d, *J* = 6 Hz, 3H, CH_3_); ^13^C NMR (CDCl_3_, 125 MHz) δ 133.9 (Ar-C); 132.7 (Ar-C), 131.4 (Ar-C), 129.2 (Ar-C), 65.6 (C-OH), 43.8 (CH_2_), 22.0 (CH_3_). The spectral data agreed with the published ^1^H-NMR [10] data.

(2*S*)-1-[(2,5-Dichlorophenyl)sulfanyl]propan-2-ol (*S*)-**1d:** White solid (42% yield), ^1^H NMR (CDCl_3_, 500 MHz) δ 7.30 (d, *J* = 2.5 Hz, 1H, Ar-H); 7.29 (s, 1H, Ar-H); 7.10 (dd, *J* = 8.5, 2.0 Hz; 1H, Ar-H); 3.98–3.92 (m, 1H, CH); 3.11 (dd, *J* = 13.5, 4 Hz, 1H, CH_2_), 2.93 (dd, *J* = 13.5, 8.5 Hz, 1H, CH_2_), 2.27 (s, 1H, OH), 1.33 (d, *J* = 6.5 Hz, 3H, CH_3_). ^13^C NMR (CDCl_3_, 125 MHz) δ 137.1 (Ar-C), 133.1 (Ar-C), 132.2 (Ar-C), 130.66 (Ar-C), 128.7 (Ar-C), 127.1 (Ar-C), 65.77 (C-OH), 42.1 (CH_2_), 22.3 (CH_3_).

(2*S*)-1-[(4-Methoxyphenyl)sulfanyl]propan-2-ol (*S*)-**1e**: Colorless liquid (50% yield), ^1^H NMR (CDCl_3_, 500 MHz) δ 7.38 (d, *J* = 9 Hz, 2H, Ar-H), 6.84 (d, *J* = 9 Hz, 2H, Ar-H), 3.79 (s, 3H, O-CH_3_), 3.78–3.73 (m, 1H, CH), 2.98 (dd, *J* = 13.5, 3.5 Hz, 1H, CH_2_), 2.72 (dd, *J* = 13.5, 8.5 Hz, 1H, CH_2_), 2.54 (s, 1H, OH), 1.22 (d, *J* = 6.5 Hz, 3H, CH_3_); ^13^C NMR (CDCl_3_, 125 MHz) δ 159.3 (Ar-C), 133.8 (Ar-C), 125.1 (Ar-C), 114.7 (Ar-C), 65.3 (C-OH), 55.4 (O-CH_3_), 45.7 (CH_2_), 21.7 (CH_3_). The spectral data agreed with the published ^1^H-NMR [10] data.

### 3.8. Bioreduction of 1-(Arylsulfanyl)propan-2-ones (***2a***–***e***) to (R)-1-(Arylsulfanyl)propan-2-ols (R)-***1a***–***e*** at Preparative Scale

LkADH whole-cell suspension (1.5 mL) was added to sodium phosphate buffer (41 mL, pH = 7, 50 mM). To the resulted suspension were added 2-propanol (2.5 mL) and the substrate (**2a**–**e**, 100 mg, in 5 mL of DMSO). The resulting mixture was shaken at 350 rpm for 24 h at room temperature. After centrifugation for 10 min at ~4000× *g*, 24 °C, the supernatant was extracted with diethyl ether (3 × 50 mL), and the combined organic layers were dried over sodium sulfate and evaporated under reduced pressure. The crude residue was purified by column chromatography over silica gel (eluent: hexane/ethyl acetate 10:1 to 7:1) to give the desired alcohols (*R*)-**1a**–**e**.

(2*R*)-1-(Phenylsulfanyl)propan-2-ol (*R*)-**1a:** Colorless liquid (73% yield). Physical data and spectra were undistiguishable from the ones of (*S*)-**1a**.(2*R*)-1-[(4-Methylphenyl)sulfanyl]propan-2-ol (*R*)-**1b**: Yellowish liquid (56% yield). Physical data and spectra were undistiguishable from the ones of (*S*)-**1b**.(2*R*)-1-[(4-Chlorophenyl)sulfanyl]propan-2-ol (*R*)-**1c**: Yellow liquid (60% yield). Physical data and spectra were undistiguishable from the ones of (*S*)-**1c**.(2*R*)-1-[(2,5-Dichlorophenyl)sulfanyl]propan-2-ol (*R*)-**1d**: White solid (62% yield). Physical data and spectra were undistiguishable from the ones of (*S*)-**1d**.(2*R*)-1-[(4-Methoxyphenyl)sulfanyl]propan-2-ol (*R*)-**1e**: Colorless liquid (52% yield). Physical data and spectra were undistiguishable from the ones of (*S*)-**1e**.

## 4. Conclusions

In this study, enantiocomplementary bioreductions of substituted 1-(arylsulfanyl)propan-2-ones **2a**–**e** were investigated using whole-cell forms of four yeast strains (WY1: *P. carsonii*, WY2: *L. elongisporus,* WY4: *C. norvegica* and WY12: *C. parapsilosis)* and two recombinant alcohol dehydrogenases (LkADH: *L. kefir*, and RaADH: *R. aetherivorans*) as biocatalysts. The bioreductions proceeded in all cases with excellent enantioselectivity (ee > 99%) and—with the exception of the less efficient WY4 and RaADH—resulted in the formation of enantiopure alcohols (either enantiomer of **1a**–**e**) with good to excellent conversions. In the bioreduction in ketone **2d**, the most pronounced steric hindrance due to the aryl group with two chlorine substituents caused a lower transformation rate (~conversion) than was observed in the other reactions. The most efficient enantiocomplementary biocatalysts were the whole-cell forms of WY12 and LkADH producing (*S*)- and (*R*)-enantiomers of the product alcohols **1a**–**e**, respectively. The bioreductions of substituted 1-(arylsulfanyl)propan-2-ones **2a**–**e** with these biocatalysts at preparative scale yielded the expected enantiopure forms of the chiral alcohols (*S*)- or (*R*)-**1a**–**e** in moderate to good yields. Thus, our study offers highly efficient methods to produce sulfur containing chiral alcohols, (*S*)-**1a**–**e** with *C. parapsilosis* (WY12) and (*R*)-**1a**–**e** with *L. kefir* alcohol-dehydrogenase-containing *E. coli* whole cells (LkADH) at preparative scale. This study with the aid of LkADH is the first report on preparative-scale (*R*)-selective bioreduction from the corresponding ketones leading to enantiopure β-hydroxysulfides. 

## Data Availability

Dataset available on request from the authors.

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
