# Peer review of "Enantiocomplementary Bioreduction of 1-(Arylsulfanyl)propan-2-ones"

_molecules, 2024, doi:10.3390/molecules29163858_

Round 1
Reviewer 1 Report
Comments and Suggestions for Authors
Authors describe the bioreduction of a set 1-(arylsulfanyl)propan-2-ones employing a set of alcohol dehydrogenases in selective bioreduction processes. Depending on the ADH used, the corresponding beta-hydroxysulfides can be achieved with high yields and optical purities, being possible to obtain both enantiomers of the final products. Best results are achieved with Witaria culture collection to get the (S)-alcohols, whereas the Lactobacillus kefir ADH afforded the opposite enantiomers. The main issue for this manuscript is that the biorduction of these compounds has been previosly performed with other ADHs (see Castagnlo and coworkers, Angew. Chem. Int. Ed. 2022, 61, e202202363). This paper, that is not cited by the authors, developed a cascade reaction in which using an initial SN2 reaction a set of b-ketosulfides are obtained, which are furhter reduced in presence of ADHs. it is true that in that paper isolated ADHs are employed, but authors studied 23 substrates with high yields and optical purities. So I would recommed to submit these results to a journal more specialized in Biocatalysis, trying to perform further research for studying some reaction parameters that could improve the work performed herein.

Author Response
Comments#1: Authors describe the bioreduction of a set 1-(arylsulfanyl)propan-2-ones employing a set of alcohol dehydrogenases in selective bioreduction processes. Depending on the ADH used, the corresponding beta-hydroxysulfides can be achieved with high yields and optical purities, being possible to obtain both enantiomers of the final products. Best results are achieved with Witaria culture collection to get the (S)-alcohols, whereas the Lactobacillus kefir ADH afforded the opposite enantiomers. The main issue for this manuscript is that the biorduction of these compounds has been previosly performed with other ADHs (see Castagnlo and coworkers, Angew. Chem. Int. Ed. 2022, 61, e202202363). This paper, that is not cited by the authors, developed a cascade reaction in which using an initial SN2 reaction a set of b-ketosulfides are obtained, which are furhter reduced in presence of ADHs. it is true that in that paper isolated ADHs are employed, but authors studied 23 substrates with high yields and optical purities. So I would recommed to submit these results to a journal more specialized in Biocatalysis, trying to perform further research for studying some reaction parameters that could improve the work performed herein.
Response #1: We thank the reviewer drawing our attention to this elegant work which we not cited (because Chemical Abstracts does not include it as direct bioreduction from the corresponding ketones). Thus, we added a section describing this work and modified the last part of the Introduction as follows:
“Notables are the elegant chemo-biocatalytic cascade reactions utilizing purified (S)- or (R)-selective KREDs and NADH supplemented reaction systems to produce the (S)- or (R)-1-(arylsulfanyl)propan-2-ols starting from 1-chloropropan-2-ol or chloroacetone and tiophenols [12].”
- F. Zhao, K. Lauder, S. Liu, J. D. Finnigan, S. B. R. Charnock, S. J. Charnock, D. Castagnolo, Angew. Chem. Int. Ed. 2022, 61, e202202363.
Although (S)- or (R)-selective KREDs were applied in this work to produce the (S)- or (R)-1-(arylsulfanyl)propan-2-ols, we should emphasize that the ee’s for the (S)-alcohols from the cascade reactions were significantly lower (~95%ee) than that achieved in our study (>99%ee). Moreover, in our whole-cell bioreductions no enzyme purification and NADH supplementation was required. Accordingly, the final sentence of the Introduction was also modified:
“Because the usefulness of various (S)-1-(arylsulfanyl)propan-2-ols were already indicated but no single step bioreductions leading to (R)-1-(arylsulfanyl)propan-2-ols from the corresponding ketones were found, our goal was to develop high efficiency methods for preparation of both enantiomeric forms of 1-(arylsulfanyl)propan-2-ols using enantiocomplementary KREDs in their whole-cell forms.”
Reviewer 2 Report
Comments and Suggestions for Authors
In this study, the author used whole cell biocatalysts to produce 1-(arylsulfanyl)propan-2-ols, the excellent conversion and enantioselectivity were obtained in the reduction of different ketones. The result is good, but many experiments need to be supplemented before it can be published.
1. The experimental details in the paper are not sufficiently described. Were the (R)-alcohol dehydrogenase and (S)-alcohol dehydrogenase been studied before? Please cite the reference in 3.2.2, not just cited in introduction. Is the alcohol dehydrogenase need to be induced to express by IPTG before cell harvested?
2. Many experimental conditions need to be optimized. Such as the amount of cell, the pH and temperature. The concentration of substrates.
3. The references are a little bit old, please replace them with some new papers.
Comments on the Quality of English LanguageThe result is good, but the experiments are too simple, more experiments are necessary to support the conclusion. Only two figures in this manuscript. Actually, figure 1 is a scheme. So, I suggest to supplement more data.
Author Response
Comment #1: In this study, the author used whole cell biocatalysts to produce 1-(arylsulfanyl)propan-2-ols, the excellent conversion and enantioselectivity were obtained in the reduction of different ketones. The result is good, but many experiments need to be supplemented before it can be published.
- The experimental details in the paper are not sufficiently described. Were the (R)-alcohol dehydrogenase and (S)-alcohol dehydrogenase been studied before? Please cite the reference in 3.2.2, not just cited in introduction. Is the alcohol dehydrogenase need to be induced to express by IPTG before cell harvested?
Response #1: We have added to Section 3.2.2 the following sentence citing the relevant references on the KREDs used: “The recombinant KREDS used in our study (the (S)-selective Rhodococcus aertherivorans alcohol dehydrogenase (RaADH)[17-20] and the (R)-selective Lactobacillus kefir alcohol dehydrogenase (LkADH)[21,22]) were comprehensively investigated on similar substrates including optimization of the bioreductions.”.
The protein expression in the recombinant E. coli cells was induced with lactose (which was a component of the carbon supplementing solution, the so called “autoinduction medium”) therefore supplementing with IPTG was not necessary to start the expression.
Comment #2: Many experimental conditions need to be optimized. Such as the amount of cell, the pH and temperature. The concentration of substrates.
Response #2: In our study, we applied the properly adopted optimized bioreduction conditions of the cited studies resulting in the biocatalytic processes as shown in Figure 1 (former Figure 2). The biotransformations with LkADH achieved the maximal 100% conversion in 2 h for 1a-c and in 24 h for 1d resulting in the corresponding (R)-alcohols with >99%ee. Similarly, biotransformations with the WY12 yeast strain produced the enantiopure (S)-alcohols (>99% ee) after 2 h with the maximal 100% conversion. Since in all cases we could achieve practically full conversion with >99%ee of the products, no further optimization was necessary.
Our results indicated which are the suitable KRED systems for obtaining enantiopure (S)- or (R)-1-(arylsulfanyl)propan-2-ols opening up the possibility of optimizations for higher substrate concentrations or lower biocatalyst-to-substrate ratio in future studies to enhance the economic viability of such biotransformations.
Comment #3: The references are a little bit old, please replace them with some new papers.
Response #3: The cited references were selected with the focus on existing data on biocatalytic production of enantiomers of chiral 1-(arylsulfanyl)propan-2-ols by kinetic resolution from the racemates or by asymmetric reductions mediated by ketoreductases from the corresponding 1-(arylsulfanyl)propan-2-ones including quite a lot of older references. As requested, we completed the introduction with recent reviews on the general biocatalytic applicability of ketoreductases by addition of the following sentence:
„Since ketoreductases are important elements of the toolbox of industrial biocatalysis [13-16], various …”
13. Li, Z.; Yang, H.; Liu, J.; Huang, Z.; Chen, F. Application of Ketoreductase in Asymmetric Synthesis of Pharmaceuticals and Bioactive Molecules: An Update (2018–2020) Chem. Rec. 2021, 21, 1611.
14. Kumari, P.; Khatik, A. G.; Patil, P. D.; Tiwari, M. S.; Nadar, S. S.; Jain, A. K. Recent immobilization techniques for ketoreductases: Its design and their industrial application. Biocatal. Agric. Biotechnol., 2024, 56, 103027.
15. Yuan, Q.; Ma, L.; Kong, W.; Liu, J.; Zhang, S.; Yan, J.; Yan, J.; Bai, J.; He, Y.; Zhou, L.; Liua, Y.; Jiang, Y. Enzymatic synthesis of chiral alcohols using ketoreductases. Catal. Rev., 2024, 1–40. https://doi.org/10.1080/01614940.2024.2313603
16. Reetz, M. T.; Qu, G.; Sun, Z. Engineered enzymes for the synthesis of pharmaceuticals and other high-value products. Nat. Synth., 2024, 3, 19–32.
Comment #4: Comments on the Quality of English Language
The result is good, but the experiments are too simple, more experiments are necessary to support the conclusion. Only two figures in this manuscript. Actually, figure 1 is a scheme. So, I suggest to supplement more data.
Response #4: Thank you for your styling observations. Figure 1 was renamed to Scheme 1, and Figure 2 to Figure 1. Since we applied yeast strains already optimized on other but similar compounds and recombinant KREDS also characterized before, no serious optimization was needed to achieve good results. The selectivities were excellent (>99% ee were observed for both the (R)- and the (S)-alcohol products) with good to excellent conversions within reasonable reaction time. Thus, we focused on reporting the results and characterize the novel chiral products. For this purpose, the one Scheme, one Figure and one Table were thought to be enough considering also the Supporting Information section with a vast amount of further data on the starting compounds and chiral products.
Reviewer 3 Report
Comments and Suggestions for Authors
Nagy, Poppe, et al present an efficient protein engineering strategy using alcohol dehydrogenases (ADHs) (also called ketoreductases, KREDS) from Lactobacillus kefir and Rhodococcus aetherivorans in order to enable the sustainable enzymatic production of certain chiral alcohols. The starting compounds (substrates) are 1-(arylsulfanyl)pro-pan-2-ones, the reduction products are chiral alcohols of pharmaceutical interest.
For true efficiency, the authors developed the use of 4 different yeast strains, which solved the problem of cofactor regeneration.
Excellent results were obtained for a whole collection of sulfur-moiety containing R- and S-alcohols. More than 99% ee was observed in all cases, with moderate to hbigh yields.
This nreviewer recommends acceptance in Molecules following minor revision along the following lines:
Since KREDS (or ADHs) play an important role in biocatalysis, the authors should mention and cite the most recent review of this field:
Reetz, Qu, & Sun, Nat. Synth. 2024, 3, 19-32.

Author Response
Comments #1: Comments and Suggestions for Authors
Nagy, Poppe, et al present an efficient protein engineering strategy using alcohol dehydrogenases (ADHs) (also called ketoreductases, KREDS) from Lactobacillus kefir and Rhodococcus aetherivorans in order to enable the sustainable enzymatic production of certain chiral alcohols. The starting compounds (substrates) are 1-(arylsulfanyl)pro-pan-2-ones, the reduction products are chiral alcohols of pharmaceutical interest.
For true efficiency, the authors developed the use of 4 different yeast strains, which solved the problem of cofactor regeneration.
Excellent results were obtained for a whole collection of sulfur-moiety containing R- and S-alcohols. More than 99% ee was observed in all cases, with moderate to hbigh yields.
This nreviewer recommends acceptance in Molecules following minor revision along the following lines:
Since KREDS (or ADHs) play an important role in biocatalysis, the authors should mention and cite the most recent review of this field:
Reetz, Qu, & Sun, Nat. Synth. 2024, 3, 19-32.
Response #1: As requested, we completed the introduction with recent reviews on the general biocatalytic applicability of ketoreductases by addition of the following sentence:
„Since ketoreductases are important elements of the toolbox of industrial biocatalysis [13-16], various …”
13. Li, Z.; Yang, H.; Liu, J.; Huang, Z.; Chen, F. Application of Ketoreductase in Asymmetric Synthesis of Pharmaceuticals and Bioactive Molecules: An Update (2018–2020) Chem. Rec. 2021, 21, 1611.
14. Kumari, P.; Khatik, A. G.; Patil, P. D.; Tiwari, M. S.; Nadar, S. S.; Jain, A. K. Recent immobilization techniques for ketoreductases: Its design and their industrial application. Biocatal. Agric. Biotechnol., 2024, 56, 103027.
15. Yuan, Q.; Ma, L.; Kong, W.; Liu, J.; Zhang, S.; Yan, J.; Yan, J.; Bai, J.; He, Y.; Zhou, L.; Liua, Y.; Jiang, Y. Enzymatic synthesis of chiral alcohols using ketoreductases. Catal. Rev., 2024, 1–40. https://doi.org/10.1080/01614940.2024.2313603
16. Reetz, M. T.; Qu, G.; Sun, Z. Engineered enzymes for the synthesis of pharmaceuticals and other high-value products. Nat. Synth., 2024, 3, 19–32.
Round 2
Reviewer 1 Report
Comments and Suggestions for Authors
As the authors have used the same substrates with a different preparation of biocatalyst, using the whole cell system, I think it is necessary to study more subsrtates in the bioreduction processes in order to be accepted for publication. If not, the difference shown regarding the previous reference is not enough.
Author Response
Comments and Suggestions for Authors
As the authors have used the same substrates with a different preparation of biocatalyst, using the whole cell system, I think it is necessary to study more subsrtates in the bioreduction processes in order to be accepted for publication. If not, the difference shown regarding the previous reference is not enough.
Response: Our study demonstrates the superior selectivity (i.e. virtually enantiopure products) for the (S)-alcohols without substituent or with a moderately bulky substituent at the position 4 of the aromatic moiety as compared to the previous reports (compare specific rotations in our study to the previously reported ones in the footnote section of Table 1).
This study is also the first direct bioreduction of the 1-(arylsulfanyl)propan-2-ones with various substituents at the aromatic moiety resulting in (R)-alcohols without the need of NADH supplementation (in most cases also higher specific rotation as reported previously, see Table 1).
We really understand that our simple production and unambiguous characterization of a wider selection of the 1-(arylsulfanyl)propan-2-ols would have further value, but the aim of this study was to demonstrate simple, generalizable and cost efficient production of virtually enantiopure (S)- and (R)-1-(arylsulfanyl)propan-2-ols without the need of external NADH supplementation.
Our fifth example, the ketone with a bulky 2,4-dichlorophenyl moiety (2d), which was never investigated before, was selected to demonstrate the generality of our methodology even with sterically demanding substrates. We predict that the bioreductions of further 5 or even 10 moderately bulky members of this series (e.g. substitutions at position 3 or 4 with F, Br, Me, Et, or even pyridyl moieties instead of phenyl) would occur smoothly and would not represent extra added value (except the precise characterization of the enantiopure products which would require double working amount).
The unambiguity of our generalizable results is demonstrated by the specific optical rotation data, since all the isolated virtually enantiopure (S)- and (R)-products (ee >99%, determined by GC) exhibited exactly opposite optical rotations which were usually significantly higher than those reported before.
Reviewer 2 Report
Comments and Suggestions for Authors
Most of the comments have been taken into consideration, but the response to comment 3 is still unacceptable. Where did the author's reaction conditions in sections 3.5-3.7 come from? If there are references, please cite them. If there are no references, the dosage of cells, pH, and substrate concentration should be optimized at least.
Author Response
Comments and Suggestions for Authors
Most of the comments have been taken into consideration, but the response to comment 3 is still unacceptable. Where did the author's reaction conditions in sections 3.5-3.7 come from? If there are references, please cite them. If there are no references, the dosage of cells, pH, and substrate concentration should be optimized at least.
Response: The conditions for bioreductions with yeast cells and recombinant LkADH and RaADH containing E. coli cells were optimized for similar bioreductions earlier. Therefore, we extended the descriptions of sections 3.5 and 3.6 with references to the previous optimizations as follows.
For Section 3.5: ”The conditions optimized in our previous work for similar bioreductions with lyophilized yeast cells [33] were applied. Thus, yeast cells …”
For Section 3.6: ”The conditions applied here were similar as optimized earlier for reaction with recombinant whole-cell LkADS [19] and RaADH [41], except the reduced amount of 2-propanol (5 V/V%) due to the larger amount of DMSO (10 V/V%) applied in the present work to enhance the substrate solubility. Thus, alcohol …”
[19]: Borzęcka, W.; Lavandera I.; Gotor, V.; Synthesis of Enantiopure Fluorohydrins Using Alcohol Dehydrogenases at High Substrate Concentrations. J. Org. Chem., 2013, 78, 7312−7317.
[41]: Eddeger, K.; Gruber, C. C.; Poessl, T. M.; Wallner, S. R.; Lavandera, I.; Faber, K.; Niehaus, F.; Eck, J.; Oehrlein, R.; Hafner, A.; Kroutil, W. Biocatalitic duterium- and hydrogen-transfer using over expressed using ADH-’A’: enhanced stereoselectivity and 2H-labeled chiral alcohols. Chem. Commun., 2006, 2402–2404.
Round 3
Reviewer 1 Report
Comments and Suggestions for Authors
Even after the comments given by the authors, the results obtained, compared with the previous shown, do not justify its publication in Molecules if further research is not performed. By this reason, the paper should be rejected.
Reviewer 2 Report
Comments and Suggestions for Authors
I have no more questions.